# RIPK2 Is Crucial for the Microglial Inflammatory Response to Bacterial Muramyl Dipeptide but Not to Lipopolysaccharide

**DOI:** 10.3390/ijms252111754

**Published:** 2024-11-01

**Authors:** Changjun Yang, Maria Carolina Machado da Silva, John Aaron Howell, Jonathan Larochelle, Lei Liu, Rachel E. Gunraj, Antônio Carlos Pinheiro de Oliveira, Eduardo Candelario-Jalil

**Affiliations:** 1Department of Neuroscience, McKnight Brain Institute, University of Florida, 1149 SW Newell Drive, Gainesville, FL 32610, USA; changjunyang@ufl.edu (C.Y.); mariacarolina.ms95@gmail.com (M.C.M.d.S.); ja.howell@ufl.edu (J.A.H.); jonlarochelle19@gmail.com (J.L.); leiliu@ufl.edu (L.L.); rgunraj@ufl.edu (R.E.G.); antoniooliveira@icb.ufmg.br (A.C.P.d.O.); 2Neuropharmacology Laboratory, Department of Pharmacology, Universidade Federal de Minas Gerais, Belo Horizonte 31270-901, Brazil

**Keywords:** inflammatory response, microglia, muramyl dipeptide, lipopolysaccharide, receptor-interacting serine/threonine protein kinase 2, Toll-like receptor, nucleotide-binding oligomerization domain-like receptor, mitogen-activated protein kinase, nuclear factor-kappa B

## Abstract

Receptor-interacting serine/threonine protein kinase 2 (RIPK2) is a kinase that is essential in modulating innate and adaptive immune responses. As a downstream signaling molecule for nucleotide-binding oligomerization domain 1 (NOD1), NOD2, and Toll-like receptors (TLRs), it is implicated in the signaling triggered by recognition of microbe-associated molecular patterns by NOD1/2 and TLRs. Upon activation of these innate immune receptors, RIPK2 mediates the release of pro-inflammatory factors by activating mitogen-activated protein kinases (MAPKs) and nuclear factor-kappa B (NF-κB). However, whether RIPK2 is essential for downstream inflammatory signaling following the activation of NOD1/2, TLRs, or both remains controversial. In this study, we examined the role of RIPK2 in NOD2- and TLR4-dependent signaling cascades following stimulation of microglial cells with bacterial muramyl dipeptide (MDP), a NOD2 agonist, or lipopolysaccharide (LPS), a TLR4 agonist. We utilized a highly specific proteolysis targeting chimera (PROTAC) molecule, GSK3728857A, and found dramatic degradation of RIPK2 in a concentration- and time-dependent manner. Importantly, the PROTAC completely abolished MDP-induced increases in iNOS and COX-2 protein levels and pro-inflammatory gene transcription of *Nos2*, *Ptgs2*, *Il-1β*, *Tnfα*, *Il6*, *Ccl2*, and *Mmp9*. However, increases in iNOS and COX-2 proteins and pro-inflammatory gene transcription induced by the TLR4 agonist, LPS, were only slightly attenuated with the GSK3728857A pretreatment. Further findings revealed that the RIPK2 PROTAC completely blocked the phosphorylation and activation of p65 NF-κB and p38 MAPK induced by MDP, but it had no effects on the phosphorylation of these two mediators triggered by LPS. Collectively, our findings strongly suggest that RIPK2 plays an essential role in the inflammatory responses of microglia to bacterial MDP but not to LPS.

## 1. Introduction

As two major forms of innate immune sensors, Toll-like receptors (TLRs) and nucleotide-binding oligomerization domain (NOD)-like receptors (NLRs) play an essential role in the recognition of pathogen-associated molecular patterns (PAMPs) to activate innate immunity and inflammatory responses against pathogenic invasion or tissue injury [1]. TLRs are transmembrane receptors that mediate bacterial recognition at the cell surface or in endosomes, where TLR4 is the most efficient member of the pattern recognition receptors (PRRs) recognized and activated by a wide variety of ligands [2]. In contrast, NLRs mediate innate immune responses through cytosolic recognition of bacterial components. NOD1 and NOD2 are two major NLRs, and they normally sense bacterial molecules produced during the synthesis and/or degradation of peptidoglycan (PGN) [3,4].

NOD1 is activated upon binding to bacterial PGN fragments containing diaminopimelic acid (DAP) [3], whereas NOD2 recognizes muramyl dipeptide (MDP) constituents [4]. Recognition of PAMPs by NOD1 and NOD2 receptors leads to the interaction between the receptor-interacting serine/threonine protein kinase 2 (RIPK2, also known as RIP2, RICK, CARDIAK, and CARD3) and these innate immune receptors through caspase recruitment domain (CARD)-CARD interaction. This is followed by the release of pro-inflammatory factors through the activation of mitogen-activated protein kinases (MAPKs), thereby leading to stimulation of transcription factor activator protein-1 (AP-1) and/or nuclear factor-kappa B (NF-κB) [5,6]. As a critical mediator modulating the pro-inflammatory signaling pathway, pharmacological inhibition, genetic deletion, or degradation using the proteolysis targeting chimera (PROTAC) of RIPK2 might be beneficial in reducing inflammation in diseases such as ischemic stroke, inflammatory bowel disease, and rheumatoid arthritis.

While several reports show that RIPK2 is critical for inflammatory signaling in peripheral immune cells, little is known about the role of RIPK2 in microglia, the main tissue-resident macrophage of the CNS. Several neurological diseases are known to be associated with microglia-driven inflammation, in which NOD2 and/or TLR4/myeloid differentiation primary response protein 88 (MyD88) signaling pathways are actively involved. For example, emerging evidence indicates that ischemic stroke induces gut permeability and enhances bacterial translocation to the ischemic brain and periphery, thus resulting in a rapid response of peripheral immune cells and activation of astroglial/microglial cells to release more pro-inflammatory factors, which could exacerbate ischemic brain injury [7,8,9,10]. In agreement with this notion, RIPK2 was found to be upregulated in a time-dependent manner during brain ischemia/reperfusion injury. Mice deficient in RIPK2 exhibited a reduced inflammatory response and neuronal apoptosis, with a smaller infarct volume and improved neurological deficits score, while neuron-specific RIPK2 overexpressing transgenic mice displayed worse stroke outcomes than wild-type (WT) controls [11].

Our recent findings also demonstrated that mice with global genetic deletion of RIPK2 or conditional deletion of microglial RIPK2 showed reduced effects on infarct volume and improved neurobehavioral outcomes after ischemic stroke compared to their WT littermates [12]. Also, NOD1 and RIPK2 were found to be increased in response to intracerebral hemorrhage (ICH) and contributed to microglial activation and the inflammatory response, while either inhibition of NOD1 with ML130 (a highly selective inhibitor of NOD1) or RIPK2 with GSK583 (a specific RIPK2 inhibitor) alleviated the ICH-induced brain damage, microglial activation, and inflammatory response in mice [13]. Treatment with a RIPK2 inhibitor at the onset of reperfusion after MCAO attenuated the RIPK2 phosphorylation induced by stroke and significantly reduced infarct volume and blood–brain barrier (BBB) disruption [14]. These findings indicate that RIPK2 is upregulated or activated in the ischemic brain and plays a detrimental role in the progression of stroke injury by increasing the neuroinflammatory response and microglial activation to stroke.

However, it still remains controversial whether the receptors NOD1/2, TLR4, or both mediate RIPK2 signaling pathways [5,6,15,16,17,18], thus resulting in inflammatory responses in pathophysiological conditions. In the present study, we have examined the role of RIPK2 in NOD2- and TLR4-dependent signaling cascades in microglial cells stimulated with MDP (a NOD2 agonist) or lipopolysaccharide (LPS, a TLR4 agonist). Our findings strongly suggest that RIPK2 plays an essential role in the inflammatory responses of microglia to the stimulation with bacterial MDP but not to LPS.

## 2. Results

### 2.1. Degradation of RIPK2 by Its Proteolysis-Targeting Chimera in a Dose- and Time-Dependent Manner

To determine whether endogenous RIPK2 is degraded by GSK3728857A, a RIPK2 proteolysis-targeting chimera (PROTAC) (Figure 1A,B), SIM-A9 mouse brain microglial cells were incubated for 4 h with various concentrations of RIPK2 PROTAC (0–10 µM) and the protein levels of RIPK2 were measured by Western blotting. As observed, RIPK2 PROTAC dose-dependently reduced the RIPK2 levels in the microglial cells (Figure 1C,D). Incubation with the optimized concentration of RIPK2 PROTAC (1 µM) for increasing amounts of time exhibited a time-dependent degradation of RIPK2 (Figure 1E,F). These results indicate that the RIPK2 PROTAC produces a robust proteolytic degradation of RIPK2 in microglia.

### 2.2. Microglia Depend on RIPK2 to Mount an Inflammatory Response to MDP

It is well documented that RIPK2 is a CARD-containing serine/threonine kinase that physically associates with the CARD of NOD1 or NOD2 through CARD-CARD interactions. In peripheral macrophages, activation of the NOD2 by its specific ligand MDP results in the recruitment of RIPK2, which will trigger the production of proinflammatory mediators through NF-κB and MAPK signaling [19]. The role of RIPK2-dependent signaling in activating the microglial inflammatory response to NOD2 or TLR4 ligands is unknown. We first investigated whether SIM-A9 microglia respond to MDP. As shown in Figure 2, incubation of microglial cells with various concentrations of MDP (0 to 10,000 ng/mL) for 24 h dose-dependently increased pro-inflammatory gene expression of *Nos2*, *Il-1β*, *Tnfα*, *Il6,* and *Mmp9*. Also, the cell viability assay did not reveal any cytotoxic effect of MDP at any concentration studied. For further experiments, we used 100 ng/mL MDP as it was the lowest concentration to induce the maximum inflammatory responses attainable with MDP in the microglial cells.

To assess the contribution of RIPK2 to NOD2- and Toll-like receptor 4 (TLR4)-mediated inflammatory responses, microglial cells were pretreated with 1 µM RIPK2 PROTAC for 4 h followed by 20 h incubation with 100 ng/mL MDP, the NOD2 agonist, or 10 ng/mL LPS, the TLR4 agonist. Of note, administration of LPS at 10 ng/mL did not display significant cytotoxicity to SIM-A9 cells at different cell densities [20]. Treatment with RIPK2 PROTAC completely abolished the MDP-induced gene expression of classical proinflammatory mediators such as *Nos2*, *Ptgs2*, *Il-1β*, *Tnfα*, *Il6*, *Ccl2,* and *Mmp9* (Figure 3A) and iNOS protein level (Figure 3B,C), which was associated with significant RIPK2 degradation (Figure 3B,E). Of interest was that RIPK2 PROTAC reduced the effects of MDP-induced *Ptgs2* mRNA expression (Figure 3A), but it had no significant effects on the COX-2 protein level (Figure 3B,D). Similar to a previous report that MDP treatment induces rapid proteasomal degradation of NOD2 and RIPK2 in RAW264.7 macrophage cells [21], our findings indicated that MDP treatment significantly reduced the RIPK2 protein level in the microglia compared to unstimulated cells (Figure 3B,E).

It is known that proinflammatory gene transcription induced by NOD1 and NOD2 is mediated through the NF-κB transcription factor and MAPK signaling pathways [19]. To determine which downstream effector was involved in MDP/NOD2-mediated inflammatory responses in microglia, phosphorylated protein levels of NF-κB p65 and p38 MAPK were measured in MDP-stimulated SIM-A9 cells with and without RIPK2 PROTAC pretreatment. As shown in Figure 4A, MDP increased phosphorylation of NF-κB p65 and p38 MAPK, and 60 min of incubation with MDP induced the maximal effects on both phosphorylated protein levels. Such activation of NF-κB p65 and p38 MAPK in response to MDP was abolished entirely in microglia in the presence of RIPK2 PROTAC (Figure 4B–D). Taken together, these findings strongly suggest that RIPK2 is essentially involved in the MDP-mediated activation of the NF-κB and p38 MAPK signaling pathways, which contribute to the proinflammatory response of microglia to MDP treatment.

### 2.3. RIPK2 Is Only Partially Required for LPS-Mediated Inflammatory Response in Microglia

Next, we examined the effects of RIPK2 PROTAC on LPS-stimulated immune inflammatory responses in the SIM-A9 cells since it remains controversial whether RIPK2 contributes to NOD1/2 signaling, TLR4 signaling, or both [5,6,15,16,17,18]. Compared with the effects of MDP on proinflammatory gene transcription, LPS has a more potent ability to stimulate inflammatory gene expression. As shown in Figure 5A–D, quantitative RT-PCR and immunoblots indicate that proinflammatory genes, including *Nos2*, *Ptgs2*, *Il-1β*, *Tnfα*, *Il6*, *Ccl2*, and *Mmp9*, as well as the protein levels of iNOS and COX-2, were dramatically upregulated by LPS in microglia. Degradation of RIPK2 by the RIPK2 PROTAC partly attenuated LPS-induced *Ptgs2*, *Il-1β*, *Il6*, *Ccl2*, and *Mmp9* gene transcription and the COX-2 protein, but it had no significant effects on the mRNA expression of *Nos2* and *Tnfα* or iNOS protein level induced by LPS. Unlike the effects mediated by MDP, increases in phosphorylation of NF-κB p65 and MAPK p38 by LPS were not attenuated with the pretreatment of the RIPK2 PROTAC (Figure 5B,E,F). Also, LPS alone robustly increased RIPK2 levels compared to the control cells (Figure 5B,G), which is in agreement with previous reports that LPS can sustainedly increase RIPK2 levels in macrophages isolated from mice or in the RAW264.7 macrophage cell line [5,21]. Collectively, these findings suggested that RIPK2 is not essential for microglia during an LPS/TLR4-mediated inflammatory response.

## 3. Discussion

In the current study, we have compared the role of RIPK2 in MDP/NOD2- and LPS/TLR4-mediated signaling pathways that essentially contribute to inflammatory responses in microglial cells. We found that PROTAC-mediated degradation of RIPK2 is concentration- and time-dependent in SIM-A9 microglia. Incubating SIM-A9 cells with MDP, the agonist of NOD2, increased pro-inflammatory gene expression of *Nos2, Il-1β, Tnfα, Il6,* and *Mmp9* in a concentration-dependent manner. Degradation of RIPK2 with the PROTAC molecule completely abolished the effects of MDP-induced iNOS and COX-2 protein levels and pro-inflammatory gene transcription of *Nos2, Ptgs2, Il-1β, Tnfα, Il6, Ccl2,* and *Mmp9*. However, increases of iNOS and COX-2 proteins, as well as seven pro-inflammatory genes induced by a TLR4 agonist, LPS, were slightly attenuated with the pretreatment of the RIPK2 PROTAC. Further findings revealed that the RIPK2 PROTAC completely blocked the activation of p65 NF-κB and p38 MAPK induced by MDP, but it had no effects on the phosphorylation of these two mediators triggered by LPS. Taken together, our findings strongly suggest that RIPK2 plays a crucial role in the inflammatory responses of microglia to bacterial MDP but not LPS.

A considerable number of studies have demonstrated that RIPK2 acts as an adaptor in immune response and inflammation process, transducing signals either from NODs (NOD1 and NOD2) or TLRs, thus leading to the activation of MAPKs and NF-κB, followed by the subsequent production of proinflammatory factors [5,6,15,16,17,18]. Several studies using RIPK2-deficient mice suggested that RIPK2 is involved in TLR signaling, such as LPS/TLR4 signaling, and the absence of RIPK2 conferred reduced inflammatory responses in macrophages stimulated with LPS [5,16,18]. In human monocyte-derived dendritic cells, knockdown of RIPK2 by siRNA blocked LPS-induced activation of p38 MAPK and NF-κB signaling, thus reducing the expression of IL-12, a critical factor for the generation of the Th1 type immune response [22]. In line with these observations, trinitrobenzene sulfonic acid (TNBS)- or dextran sodium sulfate (DSS)-induced colitis was ameliorated by administration of siRNA-targeting RIPK2 in NOD2- or NOD1/NOD2-double-deficient mice [17], indicating that the effect of RIPK2 depletion on colitis can occur independently of either NOD1 or NOD2 signaling and TLRs-dependent gut inflammation might be involved. In contrast to these reports, a study examined both innate and adaptive immune responses in *Ripk2*-deficient mice and no significant differences in IL-6 or TNFα production were observed in bone-marrow-derived macrophages from RIPK2 knockout or WT mice when stimulated with LPS (TLR4 agonist), poly(I:C) (TLR3 agonist), CpG (TLR9 agonist), or Pam3Cys-SKKKK (Pam3; synthetic peptide specific for TLR2), while NOD1/2 signaling was impaired in macrophages from *Ripk2*-deficient mice [15], indicating that RIPK2 is a downstream effector of NOD1/2 instead of TLR signaling that contributes to innate immune responses in macrophages. Consistent with this observation, Park et al. [6] demonstrated that macrophages and mice lacking RIPK2 were defective in their responses to NOD1 and NOD2 agonists but exhibited unimpaired responses to highly purified TLR ligands. Activation of NF-κB and MAPKs by LPS was greatly inhibited or abolished in TLR4-null macrophages but was unaffected in RIPK2-deficient macrophages. Such activation of NF-κB and MAPKs by MDP, the NOD2 agonist, was abrogated in *Ripk2*-deficient macrophages. Furthermore, the absence of RIPK2 or double deficiency of NOD1 and NOD2 was associated with reduced IL-6 production in *Listeria*-infected macrophages. These results suggest that RIPK2 mediates the innate immune response induced by NOD1 and NOD2, not TLRs. To explore this conflicting observation of whether RIPK2 acts as a downstream mediator of NOD1/2, TLRs, or both, we conducted studies investigating MDP/NOD2- and LPS/TLR4-mediated proinflammatory pathways in primary microglial cells in which RIPK2 was profoundly depleted by proteolytic degradation with the RIPK2 PROTAC. Notably, PROTAC-mediated protein degradation has emerged as a novel therapeutic strategy to tackle disease-causing aberrant proteins since PROTACs exhibit unprecedented efficacy and specificity in degrading target proteins compared to traditional small molecule inhibitors. Here, we employed GSK3728857A [23], a potent and effective RIPK2 PROTAC, in specifically targeting RIPK2 degradation to test the contribution of RIPK2 in the MDP/NOD2- and LPS/TLR4-mediated inflammatory pathway in primary microglial cells. Consistent with studies suggesting that RIPK2 acts as a downstream effector of NOD1/2 instead of TLR signaling [6,15], we found that degradation of RIPK2 with the PROTAC completely abolished the effects of MDP on the activation of p65 NF-κB and p38 MAPK as well as the related proinflammatory gene transcription including *Nos2*, *Ptgs2*, *Il-1β*, *Tnfα*, *Il6*, *Ccl2,* and *Mmp9*. The RIPK2 PROTAC slightly attenuated the inflammatory responses triggered by LPS, a TLR4 agonist. However, the PROTAC-mediated degradation of RIPK2 did not affect LPS-stimulated phosphorylation of p65 NF-κB and p38 MAPK, suggesting that an RIPK2-independent pathway might be involved in LPS/TLR4-induced proinflammatory responses in SIM-A9 microglial cells. However, our study cannot formally rule out the possibility that LPS stimulation may contribute directly or indirectly to NOD1/2-mediated signaling. Of particular note, PGN molecules capable of stimulating NOD1 and NOD2 are commonly present in preparations of TLR-stimulating components such as LPS preparation, and the presence of such contaminants could potentially explain the observed TLR signaling defects in RIPK2-deficient macrophages [3,24,25,26]. In agreement with this notion, Park et al. reported that deficiency of RIPK2 in macrophages or mice did not affect TLR signaling when highly purified TLR ligands were used for stimulations [6].

Although our findings strongly suggest that RIPK2 acts as a key effector in the MDP/NOD2-mediated NF-κB signaling pathway leading to the proinflammatory response of microglia, it is not known whether RIPK2 directly or indirectly activates the transcription factor and what is the potential downstream effector of the RIPK2. Emerging evidence suggests that the phosphatidylinositol 3-kinase (PI3K)/Akt pathway modulates the NF-κB signaling pathway, which regulates multiple biological processes such as cell survival, inflammation, immunity, and oncogenesis [27]. As reported, Akt mediates NF-kB activation to control T-cell growth and survival through the degradation of the NF-kB inhibitor IκB [28]. Also, Akt activated NF-kB by TNF*α* to induce immune and inflammatory responses [29]. In anti-apoptotic platelet-derived growth factor (PDGF) signaling, NF-kB was shown as a target of the PDGF/Ras/PI3K/Akt pathway to inhibit apoptosis and promote proliferation in primary fibroblasts [30]. While these findings place NF-kB as a downstream target of Akt in the sequence of signaling events, Akt can also be activated by NF-kB. In NIH3T3 fibroblast cells, overexpression of p65 NF-kB resulted in Akt phosphorylation in the absence of extracellular stimulatory factors, while elevated expression of IκB-α, a negative regulator of the NF-kB, reduced Akt phosphorylation [31,32]. Although the evidence for the connection between the PI3K/Akt and NF-κB pathways is compelling, the specific role of RIPK2 contributing to the activation of Akt or NF-kB in the inflammatory immune response is unknown. More recently, Xia et al. [33] demonstrated that mRNA expression and protein levels of RIPK2 were upregulated in myocardial ischemia/reperfusion (MI/R) rats and oxygen and glucose deprivation/reperfusion (OGD/R)-treated H9C2 cardiomyocytes. Inhibition of RIPK2 promoted cell proliferation, blocked apoptosis, reduced serum contents of inflammatory factors, and alleviated MI/R injury. Mechanistically, RIPK2 inhibition increased Akt phosphorylation while decreasing p65 NF-kB expression. Additionally, Zhao et al. [34] reported that MDP could induce Akt phosphorylation in a time- and dose-dependent manner in HEK293T cells. Pharmacological inhibition of PI3K/Akt signaling induced by the MDP enhanced the phosphorylation of p65 NF-kB on Ser529 and Ser536 residues, which resulted in the elevation of p65 transactivation activity. Based on these observations, RIPK2 might positively regulate the NF-kB signaling pathway but negatively modulate the Akt molecule to help cell survival in specific cell types such as cardiomyocytes or kidney cells. However, the precise molecular mechanisms of RIPK2-mediated neuroinflammation in microglia remain to be clarified. Future investigations will be required to explore the potential downstream effectors of the RIPK2 in the MDP/NOD2-mediated signaling pathway that triggers immune cell activation and response.

In conclusion, our findings strongly suggest that RIPK2 plays an essential role in the inflammatory responses of microglia to bacterial MDP but not LPS, and further studies will be needed to explore the molecular basis for the cooperation between the receptors of NOD2 and TLR4. Our data could have important implications for understanding the complex interplay between NOD2- and TLR4-mediated inflammatory pathways in microglia, which is relevant for many neurological diseases associated with dysregulated microglial neuroinflammatory responses.

## 4. Materials and Methods

### 4.1. Cell Culture

The SIM-A9 mouse microglial cell line was obtained from Dr. Colin K. Combs at the University of North Dakota and grown in a T75 flask with 12 mL of complete growth media supplemented with 84% DMEM/F12 media (Cat. No. SH30023.01; HyClone, Cytiva, Marlborough, MA, USA), 10% heat-inactivated fetal bovine serum (Cat. No. F4135, Millipore Sigma, St. Louis, MO, USA), 5% heat-inactivated horse serum (Cat. No. 16050130, Thermo Fisher Scientific, Waltham, MA, USA), and 1% penicillin-streptomycin (Cat. No. 15140122; Thermo Fisher Scientific) and maintained at 37 °C in a 5% CO_2_ incubator [35,36]. The cells were subcultured every 2–3 days, and passages from 5 to 25 were used in this study.

### 4.2. Treatments and Harvest of Microglial Cells

For assigned experiments, cells were seeded at a density of 3.6 × 10^4^ cells/cm^2^ in a 60-mm dish pre-coated with 50 µg/mL of poly-D-lysine (Cat. No. P6407, Millipore Sigma, St. Louis, MO, USA) with 4 mL of complete growth media and maintained at 37 °C under 5% CO_2_. After growing overnight, cells were incubated for 4 h with different concentrations (0–10 µM) of RIPK2 proteolysis-targeting chimera (PROTAC) GSK3728857A (hereafter referred to as RIPK2 PROTAC, obtained from GlaxoSmithKline under material transfer agreement number UK-JH-MTA3000037627) [23] for the dose–response experiment or varying incubation periods (0–24 h) with 1 µM of RIPK2 PROTAC for the time-course study. For optimizing the concentration of NOD2 agonist used in the following studies, cells were exposed to various concentrations (0 to 10 µg/mL) of muramyl dipeptide (MDP, a bacterial cell wall component) (Cat. No. tlrl-lmdp, InvivoGen, San Diego, CA, USA) for 24 h. After confirming the optimal concentration of MDP, cells were treated with 100 ng/mL MDP with different incubation times to determine phosphorylated protein levels involved in NF-κB and p38 MAPK signaling pathways. Also, cells were pretreated with either DMSO or the optimized concentration of RIPK2 PROTAC (1 µM) for 4 h followed by 20 h incubation of 100 ng/mL MDP or 10 ng/mL lipopolysaccharide (LPS) (Cat. No. L9641, Millipore Sigma). At the end of each incubation time, media were harvested for cell viability analysis. Cells were rinsed with ice-cold phosphate-buffered saline (PBS) and collected with 200 µL of radioimmunoprecipitation (RIPA) lysis buffer consisting of 50 mM Tris-HCl (pH 7.4), 150 mM NaCl, 5 mM EDTA, 1 mM EGTA, 1% NP-40, 0.5% sodium deoxycholate, and 0.1% SDS, plus protease and phosphatase inhibitor cocktails (Cat. Nos. 78430 and 78428, respectively; Thermo Fisher Scientific). Half of the cell lysate volume was used for RNA isolation, and the other half was used for protein extraction, as described previously [36].

### 4.3. Immunoblotting Analysis

Western blots were performed with minor modifications, as previously described [36]. Briefly, thirty micrograms of protein were denatured in 2× Laemmli sample buffer (Cat. No. 1610737, Bio-Rad, Hercules, CA, USA) containing 4% β-mercaptoethanol at 100 °C for 5 min before loading into 4–20% SDS-polyacrylamide gels. After gel running, they were transferred onto nitrocellulose membranes and then blocked for 1 h at room temperature with Intercept (TBS) Blocking Buffer (Cat. No. 927-60001, LI-COR Biotechnology, Lincoln, NE, USA). After that, the membranes were incubated at 4 °C overnight with primary antibodies rabbit anti-RIPK2 (1:1000; Cat. No. 4142; Cell Signaling Technology, Danvers, MA, USA), rabbit anti-iNOS (1:1000; Cat. No. ab15323; Abcam, Cambridge, MA, USA), rabbit anti-COX-2 (1:2000; Cat. No. ab15191; Abcam), rabbit anti-phospho-NF-κB p65 (1:1000; Cat. No. 3033; Cell Signaling Technology), rabbit anti-phospho-p38 MAPK (1:1000; Cat. No. 4631; Cell Signaling Technology), or rat anti-β-actin antibody (1:5000; Cat. No. 664801; BioLegend, San Diego, CA, USA) in Intercept T20 (TBS) Antibody Diluent (CAT#: 927-65001). The membranes were then washed with TBST three times at 5 min intervals, incubated with goat anti-rabbit IRDye 800CW (1:30,000; Li-Cor, Lincoln, NE, USA) or goat anti-rat IRDye 680LT (1:40,000; Li-Cor) secondary antibodies for 1 h at room temperature. Immunoreactive bands were visualized and densitometrically analyzed using Odyssey infrared scanner and Image Studio 2.0 software (Li-Cor).

### 4.4. RNA Extraction

The remaining half of the cell lysate was mixed with 1.0 mL of TRIzol solution (Cat. No. 15596026; Thermo Fisher Scientific) to be processed for RNA isolation. Briefly, 200 μL of chloroform was added to the mixture of cell lysates/TRIzol and vortexed well, incubated at room temperature for 3 min, then centrifuged at 12,000× *g* for 15 min at 4 °C. The upper aqueous phase was saved, mixed with an equal volume of 70% ethanol, and vortexed thoroughly before filtering the mixture on a silica column (Cat. No. SD5008; Bio Basic) at 12,000× *g* for 1 min at 4 °C. After filtration, the nucleic acids bound to the silica resin of the column were washed with 250 μL of washing buffer A (1.0 M Guanidine thiocyanate, 10 mM Tris, pH 7.0) and spun down at 12,000× *g* for 1 min. For DNA removal, 75 μL of DNase I digestion mixture (66 µL nuclease-free water + 1.5 µL DNase I (Cat. No. 6344; Worthington Biochemical Corporation, Lakewood, NJ, USA) + 7.5 µL of 10X NEB DNase I Reaction Buffer (Cat. No. B0303S; New England BioLabs^®^ Inc., Ipswich, MA, USA) was added to the silica resin bed of the column and incubated at room temperature for 15 min. The column was washed again by adding 250 μL of washing buffer A and spun down at 12,000× *g* for 1 min. The filtrate was discarded and the silica column bound with RNA sample was washed with 500 μL of washing buffer B (10 mM Tris, pH 7.0 in 80% ethanol) and spun down at 12,000× *g* for 1 min. After that, the RNA sample was washed with washing buffer B again and spun down a final time at 16,000× *g* for 2 min. Then, the silica column was transferred to a new 1.5 mL nuclease-free microcentrifuge tube, and 50 μL of nuclease-free water (pre-warmed at 70 °C) was added to the center of the column and incubated at room temperature for 3 min. The RNA was eluted by spinning at 16,000× *g* for 2 min. The RNA concentration and purity were measured by absorbance (260/280 ratio) using a Take3 Micro-Volume Plate Reader (Biotek Instruments, Winooski, VT, USA).

### 4.5. Real-Time qPCR

Real-time quantitative PCR reactions were conducted in a 96-well plate using a CFX96 Touch Real-Time PCR System (Bio-Rad). Each reaction was performed in a 10 μL volume containing 1× Luna Universal One-Step Reaction Mix plus 1× Luna WarmStart^®^ RT Enzyme Mix (Cat. No. E3005; New England BioLabs^®^ Inc.), 0.4 µM of each primer and 2 µL of 10 ng/µL RNA, using the following thermal conditions: 55 °C for 10 min, 95 °C for 1 min, followed by 40 cycles of 95 °C for 10 s and 60 °C for 30 s. A melting curve analysis (60 °C to 95 °C) was performed at the end of each PCR to further confirm the amplicons’ specificity. Primer sequences for Nos2, Ptgs2, Il-1β, Tnfα, Il6, Ccl2, Mmp9, and housekeeping genes Cyc1 and Rltr2aiap are included in Table 1. Each sample was run in duplicate, and cycle threshold (Ct) values were normalized to the housekeeping genes *Cyc1* and *Rltr2aiap* (one of the expressed repetitive elements) since they are universally stable and widely used as reference genes under various conditions [37,38].

### 4.6. Statistical Analysis

Data were expressed as mean ± SEM from at least three independent experiments and were analyzed using GraphPad Prism version 8 (GraphPad Software, San Diego, CA, USA) with a one-way ANOVA followed by Bonferroni post hoc test. A value of *p* < 0.05 was considered statistically significant.

## Figures and Tables

**Figure 1 ijms-25-11754-f001:**
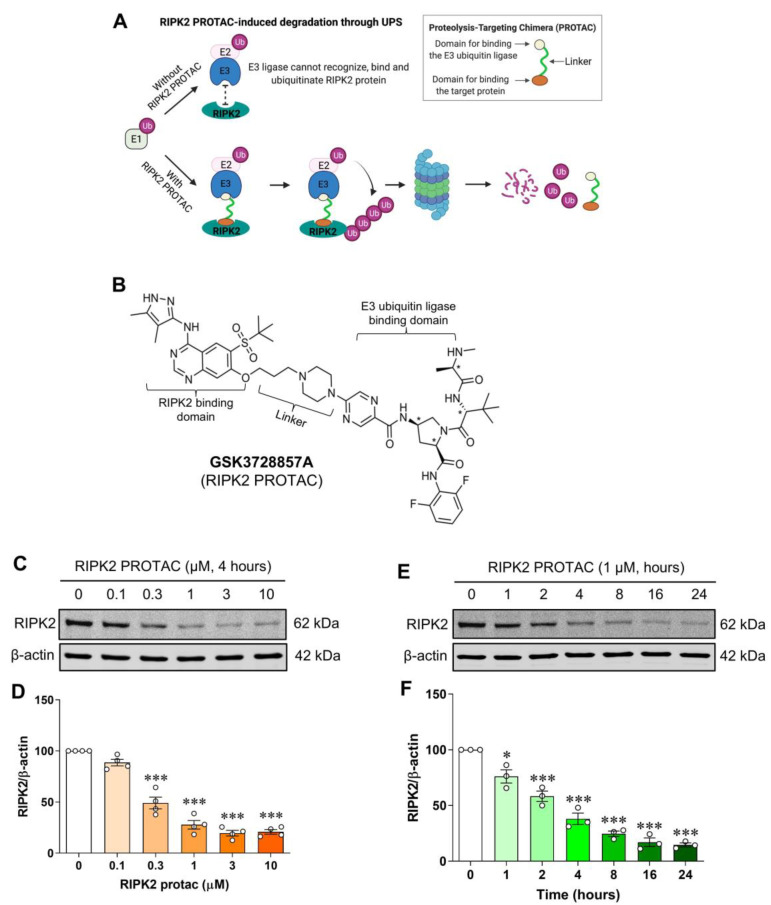
Dose- and time-dependent degradation of RIPK2 by its proteolysis-targeting chimera in SIM-A9 cells. Model of RIPK2 degradation mediated by its proteolysis-targeting chimera (PROTAC) molecule GSK3728857A (**A**) and the molecular structure of the RIPK2 PROTAC (**B**). Representative immunoblots and graphs showing degradation of RIPK2 by RIPK2 PROTAC in microglial cells. (**C**,**D**) Incubation with various concentrations of RIPK2 PROTAC (0–10 μM) for 4 h degrades RIPK2 in a dose-dependent manner. (**E**,**F**) Similarly, time-dependent degradation of RIPK2 by RIPK2 PROTAC (1 μM) was also observed. One-way ANOVA with Bonferroni post-test; * *p* < 0.05 and *** *p* < 0.001 compared with control conditions. Data are normalized to β-actin and represented as mean ± SEM from three to four independent experiments.

**Figure 2 ijms-25-11754-f002:**
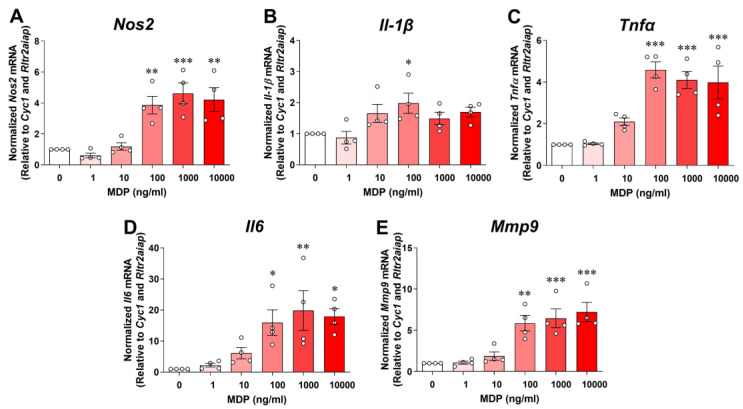
MDP dose-dependently induces pro-inflammatory gene expression in SIM-A9 microglial cells. Graphs show treatment with various concentrations (0 to 10,000 ng/mL) of muramyl dipeptide (MDP) for 24 h dose-dependently increases transcription of pro-inflammatory genes *Nos2* (**A**), *Il-1β* (**B**), *Tnfα* (**C**), *Il6* (**D**) and *Mmp9* (**E**) in microglial cells. qRT-PCR data are normalized to reference genes *Cyc1 and Rltr2aiap* and represented as fold increases compared to control cells. One-way ANOVA with Bonferroni post-test; * *p* < 0.05, ** *p* < 0.01, and *** *p* < 0.001 compared with control conditions. Data are represented as mean ± SEM from four independent experiments.

**Figure 3 ijms-25-11754-f003:**
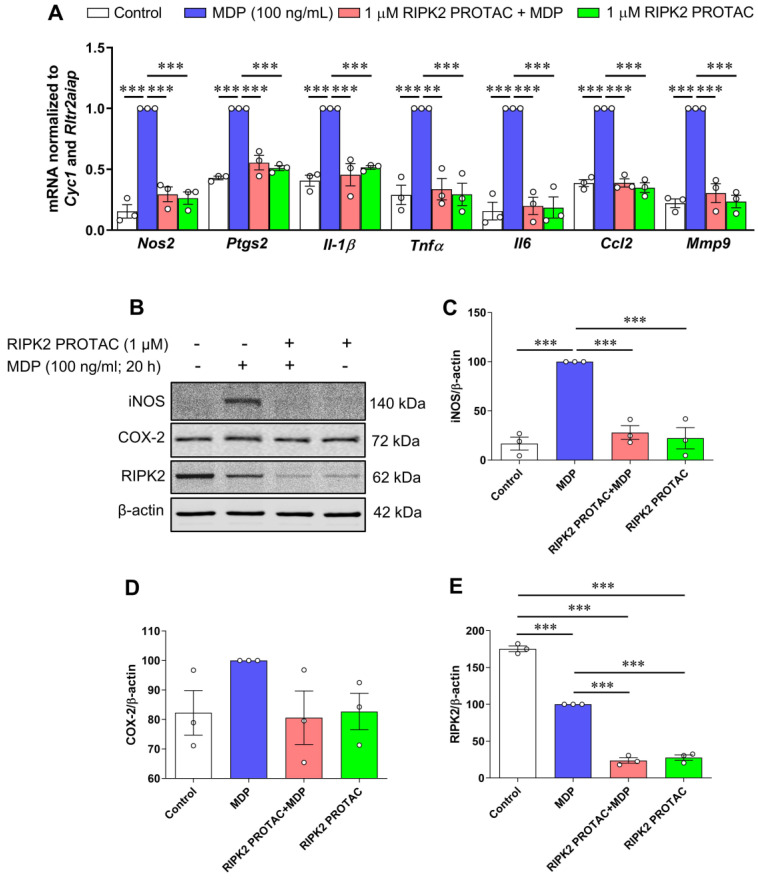
RIPK2 PROTAC reduces MDP-induced pro-inflammatory gene expression and iNOS protein levels in SIM-A9 cells. SIM-A9 cells were pretreated with 1 µM RIPK2 PROTAC for 4 h followed by 20 h incubation with 100 ng/mL MDP. After that, cells were harvested for RNA and protein extraction. (**A**) Graph shows RIPK2 degradation by RIPK2 PROTAC completely reduced MDP-induced transcription of pro-inflammatory genes *Nos2*, *Ptgs2*, *Il-1β*, *Tnfα*, *Il6*, *Ccl2,* and *Mmp9*. (**B**–**E**) Effects of RIPK2 PROTAC on MDP-induced iNOS, COX-2, and RIPK2 protein levels. qRT-PCR data are normalized to reference genes *Cyc1 and Rltr2aiap* and represented as fold changes compared to MDP treatment, and immunoblot data are normalized to β-actin. One-way ANOVA with Bonferroni post-test; *** *p* < 0.001. Data are represented as mean ± SEM from three independent experiments.

**Figure 4 ijms-25-11754-f004:**
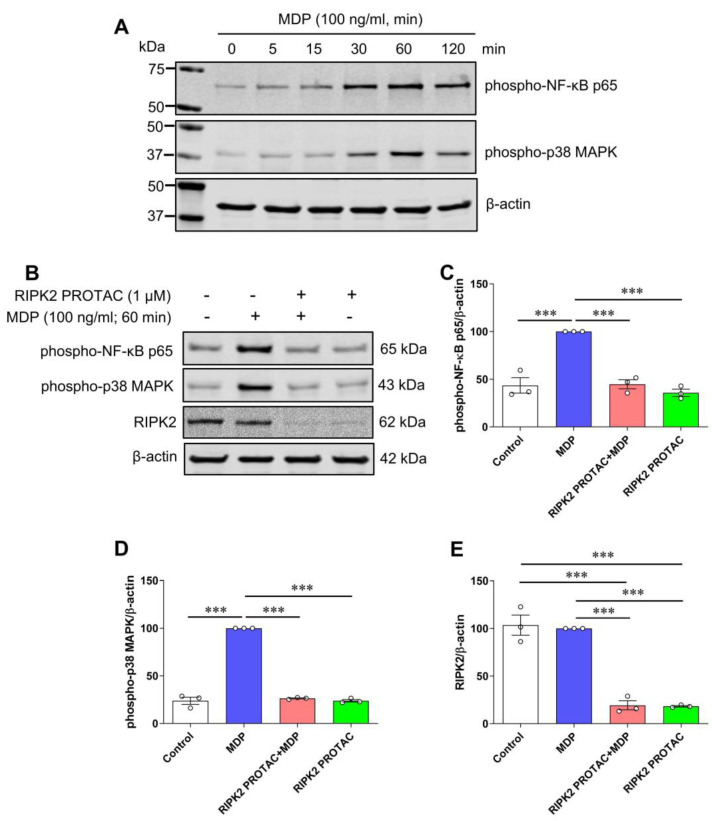
RIPK2 PROTAC suppresses activation of both NF-κB p65 and MAPK p38 induced by MDP in SIM-A9 cells. (**A**) SIM-A9 cells were stimulated with 100 ng/mL MDP for indicated periods (0 to 120 min). Immunoblots show MDP increased phosphorylation of NF-κB p65 and MAPK p38, and 60 min incubation of MDP induced maximal effects on both phosphorylated protein levels. Data are representative of three independent experiments with similar results. (**B**–**E**) SIM-A9 cells were pretreated with 1 µM RIPK2 PROTAC for 4 h followed by 60 min incubation of 100 ng/mL MDP. After that, cells were harvested for protein extraction and Western blots. Immunoblots and graphs show that RIPK2 PROTAC completely abolished effects of MDP on phosphorylation of both NF-κB p65 (**B**,**C**) and MAPK p38 (**B**,**D**), which was associated with marked degradation of RIPK2 by its PROTAC pretreatment (**B**,**E**). One-way ANOVA with Bonferroni post-test; *** *p* < 0.001. Data are normalized to β-actin and represented as mean ± SEM from three independent experiments.

**Figure 5 ijms-25-11754-f005:**
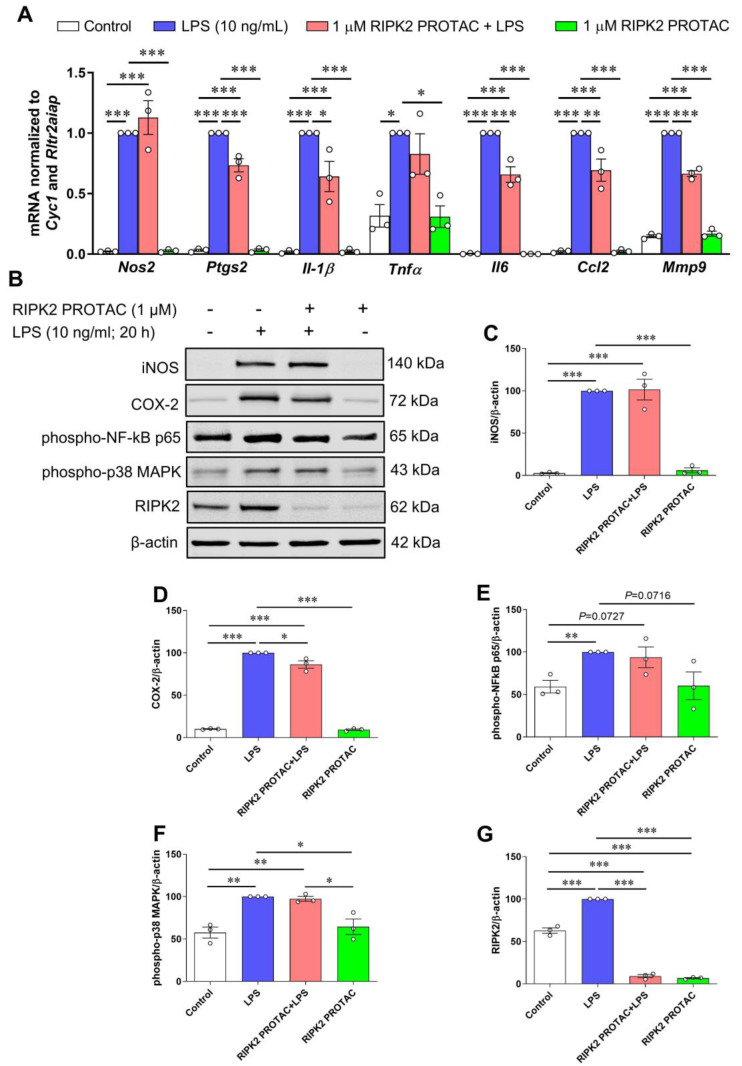
Effects of RIPK2 PROTAC on LPS-induced pro-inflammatory gene expression, protein levels of iNOS and COX-2, and phosphorylated NF-κB p65 and MAPK p38 levels in microglia. SIM-A9 microglial cells were pretreated with 1 µM RIPK2 PROTAC for 4 h followed by 20 h incubation with 10 ng/mL lipopolysaccharide (LPS). After that, cells were harvested for RNA and protein extraction. (**A**) Graph shows that RIPK2 PROTAC partly reduced LPS-induced *Ptgs2*, *Il-1β*, *Il6*, *Ccl2,* and *Mmp9* gene transcription but did not affect gene expression of *Nos2* and *Tnfα*. (**B**–**E**) RIPK2 PROTAC had no effects on LPS-induced increase in iNOS protein levels (**B**,**C**), but it slightly attenuated LPS-induced upregulation of COX-2 levels (**B**,**D**). Treatment with RIPK2 PROTAC had no effects on increased levels of phosphorylated NF-κB p65 (**B**,**E**) or p38 MAPK (**B**,**F**) induced by LPS. At same time, LPS-triggered upregulation of RIPK2 was potently degraded by its PROTAC (**B**,**G**). qRT-PCR data are normalized to reference genes *Cyc1 and Rltr2aiap* and represented as fold changes compared to LPS treatment, and immunoblot data are normalized to β-actin. One-way ANOVA with Bonferroni post-test; * *p* < 0.05, ** *p* < 0.01, and *** *p* < 0.001. Data are represented as mean ± SEM from three independent experiments.

**Table 1 ijms-25-11754-t001:** Primer sequences used for real-time PCR.

Gene	Accession Number	Forward	Reverse
*Nos2*	NM_010927	5′-GTTCTCAGCCCAACAATACAAGA-3′	5′-GTGGACGGGTCGATGTCAC-3′
*Ptgs2*	NM_011198	5′-CAAGACAGATCATAAGCGAGGA-3′	5′-GCGCAGTTTATGTTGTCTGTC-3′
*Il-1β*	NM_008361	5′-GACCTGTTCTTTGAAGTTGACG-3′	5′-CTCTTGTTGATGTGCTGCTG -3′
*Tnfα*	NM_013693	5′-AGACCCTCACACTCAGATCA-3′	5′-TCTTTGAGATCCATGCCGTTG-3′
*Il-6*	NM_031168	5′-AGCCAGAGTCCTTCAGAGA-3′	5′-TCCTTAGCCACTCCTTCTGT -3′
*Ccl2*	NM_011333	5′-CATCCACGTGTTGGCTCA-3′	5′-AACTACAGCTTCTTTGGGACA-3′
*Mmp9*	NM_013599	5′-GACATAGACGGCATCCAGTATC-3′	5′-GTGGGAGGTATAGTGGGACA-3′
*Cyc1*	NM_025567	5′-CCAAAACCATACCCTAACCCT-3′	5′-CTGCTCACTGGCTACTGTG-3′
*Rltr2aiap*	N/A	5′-CATGTGCCAAGGGTAGTTCTC-3′	5′-GCAAGAGAGAGAGAATGGCGAAAC-3′

## Data Availability

The raw data supporting the conclusions of this article will be made available by the authors upon request.

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
