# Peer review of "RIPK2 Is Crucial for the Microglial Inflammatory Response to Bacterial Muramyl Dipeptide but Not to Lipopolysaccharide"

_ijms, 2024, doi:10.3390/ijms252111754_

Round 1
Reviewer 1 Report
Comments and Suggestions for Authors
This paper discusses the role of RIPK2 (Receptor-Interacting Serine/Threonine Protein Kinase 2) in the inflammatory response of microglia, specifically in response to bacterial peptidoglycan, but with a weaker response to lipopolysaccharide (LPS). The study demonstrates that RIPK2 plays a crucial role in NOD2 (Nucleotide-binding Oligomerization Domain Receptor)-mediated signaling, but is not essential for TLR4 (Toll-Like Receptor 4)-mediated signaling.
The research used a highly specific PROTAC molecule (GSK3728857A) to degrade RIPK2, finding that it completely blocked the inflammatory response of microglia induced by bacterial peptidoglycan (MDP), such as the expression of inducible nitric oxide synthase (iNOS) and cyclooxygenase-2 (COX-2). However, PROTAC only slightly inhibited the inflammatory response induced by LPS, indicating that RIPK2 is more critical for peptidoglycan-mediated responses rather than LPS-mediated ones.
This study reveals the important role of RIPK2 in the response of microglia to bacterial infection, particularly through the NOD2 receptor-mediated pathway, while its role in TLR4-mediated inflammatory responses is relatively weaker. I have two question for this paper:
-
In my personal opinion, layout is a fundamental aspect. I hope that in the future, the layout will avoid having one page for text and another for figures. Please reformat the layout accordingly.
-
The discussion should explore the potential downstream pathways: RIPK2 triggers an inflammatory response by activating NF-κB and MAPK signaling pathways, while AKT is also involved in regulating these pathways. AKT often regulates cell survival, inflammation, and metabolic activities through the PI3K (Phosphatidylinositol 3-Kinase) pathway. Research suggests that AKT can regulate NF-κB activity through various pathways, thus participating in the inflammatory process. Therefore, the inflammatory signals triggered by RIPK2 via NF-κB may intersect with the AKT-regulated cell survival and inflammatory responses. The downstream effects of RIPK2 could indirectly influence the PI3K/AKT pathway. The PI3K/AKT pathway, along with the inflammation induced by RIPK2, may jointly regulate immune cell activation and response. For instance, in the inflammatory microenvironment, the PI3K/AKT pathway can regulate cell metabolism to support the inflammatory response, and even tumors like SCC (J Han, IJBS, 2021) (Haiyan Chen, Theranostics, 2019). Meanwhile, RIPK2 can enhance inflammatory signaling, suggesting potential functional synergy between the two. I hope some interesting downstream signaling can be added to the discussion.
Overall, I appreciate the expression in this paper; there is substantial data, but to explore infinite possibilities within the limited literature, we need to think and discuss more.
Author Response
Reviewer #1 (Comments to the Authors)
Comments 1: In my personal opinion, layout is a fundamental aspect. I hope that in the future, the layout will avoid having one page for text and another for figures. Please reformat the layout accordingly.
Response 1: Thank you for this suggestion. We have re-formatted the revised manuscript accordingly.
Comments 2: The discussion should explore the potential downstream pathways: RIPK2 triggers an inflammatory response by activating NF-κB and MAPK signaling pathways, while AKT is also involved in regulating these pathways. AKT often regulates cell survival, inflammation, and metabolic activities through the PI3K (Phosphatidylinositol 3-Kinase) pathway. Research suggests that AKT can regulate NF-κB activity through various pathways, thus participating in the inflammatory process. Therefore, the inflammatory signals triggered by RIPK2 via NF-κB may intersect with the AKT-regulated cell survival and inflammatory responses. The downstream effects of RIPK2 could indirectly influence the PI3K/AKT pathway. The PI3K/AKT pathway, along with the inflammation induced by RIPK2, may jointly regulate immune cell activation and response. For instance, in the inflammatory microenvironment, the PI3K/AKT pathway can regulate cell metabolism to support the inflammatory response, and even tumors like SCC (J Han, IJBS, 2021) (Haiyan Chen, Theranostics, 2019). Meanwhile, RIPK2 can enhance inflammatory signaling, suggesting potential functional synergy between the two. I hope some interesting downstream signaling can be added to the discussion.
Response 2: Thank you for your constructive comments and suggestions. There are so many signaling pathways, such as PI3K/Akt, MAPK, JAK-STAT, TGF-β, Wnt, and Notch, that have been reported to have crosstalk with NF-kB signaling (Guo et al. Signal Transduct Target Ther, 2024). It is really challenging to rule out which downstream effector of the RIPK2 is involved in activating p65 NF-κB and p38 MAPK. Following your suggestions, we have discussed the potential interplay of RIPK2, Akt, and NF-kB signaling pathways based on updated publications, as highlighted in red in the Discussion section.

Reviewer 2 Report
Comments and Suggestions for Authors
Journal: IJMS (ISSN 1422-0067)
Manuscript ID: ijms-3277179
Type:Article
The manuscript Title “RIPK2 is crucial for the microglial inflammatory response to bacterial muramyl dipeptide but not to lipopolysaccharide” by Changjun Yang et al, investigates a crucial area of neuropharmacology by focusing on RIPK2, an essential component of responses to inflammation. The research is extremely useful for neuroinflammation therapy techniques because of the innovative component added by the use of PROTAC technology.
Enhancing the effect and repeatability of the study can be achieved by making sure all experimental steps are fully justified and by making the data presentation clearer.
To address these important concerns, I suggest minor review.
Abstract:
1. Line no. 11: “Abstract: and eceptor-interacting serine/threonine protein kinase (RIPK2) is a kinase that plays”, the first sentence has sentence error, please correct it.
2. Line no. 26: pro-inflammatory gene transcription induced by the TLR4 agonist, LPS, were only slightly, grammatical error.
Introduction:
3. Provide precise function of TLR4-LPS signaling in microglial cells, particularly when compared to MDP-NOD2 signaling?
4. Describe the significance of microglial responses in relation to RIPK2 signaling?
Materials and methods
Cell culture
5. Add the media preparation method precisely.
6. As it is mentioned cells were subcultured every 2-3 days but did not mentioned the passage number since the behavior and responses of the cells could be impacted by more passage numbers.
Treatments and harvest of microglial cells
7. It indicated that 10 ng/mL of LPS but 100 ng/mL of MDP were applied to the cells. It would be helpful to clarify why these agonists are utilized at various concentrations and whether optimization was done for each.
8. How were the qPCR primers validated in terms of specificity and efficiency?
9. Figure 1: A. Clarify how E3 ligase interact with the target protein and why this specific E3 ligase is selected?
10. What parameters were used to choose which genes (such Nos2, Il1-β, and T nfα) to examine in relation to RIPK2 PROTAC and MDP treatment? Were any other genes evaluated that weren't displayed?
11. Have additional cellular pathways examined, such as those implicated in autophagic or apoptosis processes, to verify that the effects seen are exclusively attributable to RIPK2 degradation?
Note: Fix all the grammatical error in the manuscript.
Reduce the percentage match (plagiarism)
Cite the all experimental protocol. (Immunoblotting analysis, RNA extraction)
Some sentences are too long without reference, add the appropriate references specially in discussion part.
Author Response
Reviewer #2 (Comments to the Authors)
Comment 1: Line no. 11: “Abstract: and eceptor-interacting serine/threonine protein kinase (RIPK2) is a kinase that plays”, the first sentence has sentence error, please correct it.
Response 1: Thank you for noticing this error. This correction was made.
Comments 2: Line no. 26: pro-inflammatory gene transcription induced by the TLR4 agonist, LPS, were only slightly, grammatical error.
Response 2: To the best of our knowledge, the entire sentence is “However, increases in iNOS and COX-2 proteins and pro-inflammatory gene transcription induced by the TLR4 agonist, LPS, were only slightly attenuated with the GSK3728857A pretreatment.” Thus, “were” instead of “was” should be used.
Comments 3: Provide precise function of TLR4-LPS signaling in microglial cells, particularly when compared to MDP-NOD2 signaling?
Response 3: In microglial cells, LPS/TLR4 signaling acts as a crucial pathway to detect the presence of bacterial infection, primarily from Gram-negative bacteria, by triggering an inflammatory response through the activation of the TLR4 upon binding to the LPS, leading to the release of pro-inflammatory factors and initiating immune responses within the central nervous system (CNS). However, MDP/NOD2 signaling acts as a key pathway to sense the bacteria infection, specifically by recognizing a bacterial cell wall component called muramyl dipeptide (MDP), activating the inflammatory response through the NOD2 receptor, thus resulting in the activation of microglia and the production of inflammatory mediators, which could help fight bacterial infections in the CNS.
Comments 4: Describe the significance of microglial responses in relation to RIPK2 signaling?
Response 4: RIPK2 plays a crucial role in the inflammatory response of microglia that helps the brain from infection, chemical insult, or necrosis. RIPK2 can be used as a potential therapeutic target for neurologic disorders related to inflammation.
Comments 5: Add the media preparation method precisely.
Response 5: Thank you for this suggestion. This was updated in the “Materials and Method” section of 4.1. “Cell culture” is highlighted in red font.
Comments 6: As it is mentioned cells were subcultured every 2-3 days but did not mentioned the passage number since the behavior and responses of the cells could be impacted by more passage numbers.
Response 6: Thank you for this suggestion. This was updated in the “Materials and Method” section of 4.1. “Cell culture” is highlighted in red font.
Comments 7: It indicated that 10 ng/mL of LPS but 100 ng/mL of MDP were applied to the cells. It would be helpful to clarify why these agonists are utilized at various concentrations and whether optimization was done for each.
Response 7: Thank you for asking us to clarify why different doses of bacterial LPS and MDP were used in this study. LPS is an inflammatory stimulator widely used in in vitro cell cultures, where it can drive microglial activation through the TLR4 signaling pathway triggering pro-inflammatory gene transcription. Yousef et al’s 2022 paper (Yousef et al. Mol Neurobiol, 2022) determined the safe concentration for SIM-A9 cells with 24 hours incubation of LPS at 10, 100, and 1000 ng/mL with different cell density ranging from 0.5 × 105 cells/ml to 5 × 105 cells/ml. They found that LPS at 10 ng/mL was not associated with significant cytotoxicity across all cell densities. Based on these findings, 10 ng/mL LPS was used in our study to stimulate SIM-A9 cells to avoid any cell cytotoxicity, and this paper has been cited as reference 20 in our revised manuscript. Compared to the LPS treatment, our dose-dependent experiment findings optimized 100 ng/mL of MDP, as demonstrated in the Result section 2.2 and data shown in Figure 2.
Comments 8: How were the qPCR primers validated in terms of specificity and efficiency?
Response 8: As described in the Method section 4.5 in the manuscript, a melting curve analysis (60 °C to 95 °C) was performed at the end of each qPCR, and a single peak was observed, confirming the specificity of primers using the Bio-Rad CFX Manager software (Version 3.0.1224). With the analysis by the software, the primers’ efficiencies were achieved between 98-100%, and the R2 of the standard curve was more than 0.980.
Comments 9: Figure 1: A. Clarify how E3 ligase interact with the target protein and why this specific E3 ligase is selected?
Response 9: As reported by Mares et al (Mares et al, Communications Biology, 2020), GSK pharmaceutical company developed and identified a series of optimized PROTACs that is able to degrade the RIPK2 kinase in vivo, and the GSK3728857A (hereafter referred to as RIPK2 PROTAC) is one of them. The RIPK2 PROTAC is a bifunctional molecule that consists of a RIPK2 binding domain, a chemical linker, and an E3 ubiquitin ligase binding domain, as shown in Fig. 1B. The RIPK2 PROTAC specifically brings the E3 ligase into close proximity to the RIPK2 kinase and subsequently catalyzes the transfer of ubiquitin from the E2 to the RIPK2. The RIPK2 is tagged with ubiquitin and degraded by the ubiquitin-proteasome system (UPS). This is described in the revised manuscript and depicted in Figure 1.
Comments 10: What parameters were used to choose which genes (such Nos2, Il1-β, and Tnfα) to examine in relation to RIPK2 PROTAC and MDP treatment? Were any other genes evaluated that weren't displayed?
Response 10: In this study, a total of seven inflammation-related genes, including Nos2, Ptgs2, Il-1β, Tnfα, Il6, Ccl2, and Mmp9 were measured, and these cytokines (Il-1β, Tnfα, Il6), chemokines (Ccl2), and enzymes (Nos2, Ptgs2, Mmp9) have been well reported to be upregulated in the ischemic brain, most likely in microglia (Yang et al. Am J Physiol Cell Physiol. 2019). In our previous study (DeMars et al. Biochem Biophys Res Commun. 2018), we confirmed the upregulation of these seven genes in the mouse microglial cell lines, SIM-A9 cells, induced by bacterial LPS. To compare the role of the RIPK2 in LPS/TLR4- and MDP/NOD2-mediated signaling pathways in microglial cells, it is thus relevant to measure these pro-inflammatory genes.
Comments 11: Have additional cellular pathways examined, such as those implicated in autophagic or apoptosis processes, to verify that the effects seen are exclusively attributable to RIPK2 degradation?
Response 11: We appreciate your helpful comments, and we acknowledge that RIPK2 is implicated in contributing to multiple cellular pathways, but unfortunately, we have only focused on comparing the role of the RIPK2 in LPS/TLR4- and MDP/NOD2-mediated signaling pathways in this study. We did add one new paragraph to discuss the potential interplay of RIPK2, Akt, and NF-kB signaling pathways based on updated publications as highlighted in red font in the Discussion section. We will definitely move forward and investigate the potential role of RIPK2 in different cellular pathways in the future.

Round 2
Reviewer 1 Report
Comments and Suggestions for Authors
I think it's ready for publication.